# Endothelial Slit2 guides the Robo1-positive sympathetic innervation during heart development

Juanjuan Zhao[1], Susann Bruche[1,2], Konstantinos Lekkos[1,2], Carolyn Carr[1], Joaquim Miguel Vieira[3], John Parnavelas[4], William D Andrews[4], Mathilda Mommersteeg[1,2]*

[1]Department of Physiology, Anatomy and Genetics, University of Oxford, Oxford, United Kingdom; [2]Institute of Developmental and Regenerative Medicine, Old Road Campus, University of Oxford, Oxford, United Kingdom; [3]School of Cardiovascular and Metabolic Medicine and Sciences, Faculty of Life Sciences and Medicine, Kings College London, British Heart Foundation Centre – James Black Centre, London, United Kingdom; [4]Department of Cell and Developmental Biology, University College London, London, United Kingdom

## eLife Assessment

This study presents **important** findings on the role of Slit-Robo signaling in cardiac innervation. The evidence supporting the main claims of the authors is **convincing**. The use of several mouse models including constitutive and cell type specific knockout models make the findings more robust. The scope of the presented studies is fitting, as they primarily focus on evaluating the phenotypic changes in cardiac innervation following the loss of various Slit or Robo genes

**\*For correspondence:**
mathilda.mommersteeg@dpag.ox.ac.uk

**Competing interest:** The authors declare that no competing interests exist.

**Abstract** Axon guidance cues direct nerves in the heart during development, disease, and regeneration. These cues determine cardiac innervation patterning by regulating the balance between chemo-attraction and chemo-repulsion. However, the role of one of the most crucial ligand-receptor combinations among axon guidance molecules, the Slit chemo-active ligands and their Roundabout (Robo) transmembrane receptors, remains unknown during cardiac innervation patterning. To test if Slit-Robo signalling is important for cardiac innervation guidance, we analysed Slit and Robo mouse knock-outs. Constitutive *Slit2*[-/-] ventricles showed significantly reduced innervation, while *Slit3*[-/-] hearts showed temporary increased levels of innervation compared to wild-type littermate controls. Whereas innervation was not affected in *Robo2*[-/-] mice, the phenotype seen in *Slit2*[-/-] ventricles was recapitulated in *Robo1*[-/-] mice. Detailed expression analysis identified expression of Slit2 ligand in the endothelium of the coronary vessels, while Slit3 was highly present in the coronary smooth muscle wall and in the innervation. Both the Robo1 and Robo2 receptors were present in the nerves and at low levels in the vessels. Knocking out *Slit2* specifically in the endothelium recapitulated the defects seen in the constitutive *Slit2*[-/-] hearts. Ex vivo axon guidance cultures showed that attraction of axons extending from the ganglia was strongly reduced in ventricles with absence of endothelial Slit2 compared to wild-type controls. In the absence of endothelial Slit2, adult mice showed reduced response to challenging the sympathetic innervation. In conclusion, we have identified an important new chemo-active Slit2-Robo1 pathway required for correct cardiac innervation development.

## Introduction

A number of recent studies have revealed the extent and importance of the innervation of the heart during disease, regeneration, and aging (*Mangoni and Nargeot, 2008*; *Ieda and Fukuda, 2009*; *Crick et al., 1994*; *Salamon et al., 2023*; *Wagner et al., 2023*). Alterations in the innervation play a role in the susceptibility to arrhythmias in various cardiac diseases, from cardiac ganglia in the pathogenesis of atrial fibrillation to ventricular nerves in the predilection toward ventricular arrhythmias and sudden cardiac death in myocardial scar areas, chronic congestive heart failure, diabetic neuropathy, and diseases such as Long QT and Brugada syndrome (*Ieda and Fukuda, 2009*; *Kapa et al., 2010*; *Chen et al., 2001*; *Chen et al., 2014*; *Florea and Cohn, 2014*; *Gardner et al., 2015*; *Vaseghi et al., 2012*).

Previous studies have shown that neurogenic genes, including Semaphorin3a (Sema3a) and its receptor Neuropilin1 (Nrp1), Nerve Growth Factor (NGF), glial cell line-derived neurotrophic factor (GDNF), Edn1, and its receptor Ednra, as well as Plexin A4 are involved in determining cardiac innervation patterning during development by regulating the balance between chemo-attraction and chemo-repulsion (*Miwa et al., 2013*; *Kimura et al., 2012*; *Maden et al., 2012*; *Ieda et al., 2006*; *Kuruvilla et al., 2004*; *Poltavski et al., 2019*; *Ieda et al., 2007*). Some of these pathways are reactivated during aging as accumulation of senescent cells stimulates the release of Sema3a from the coronary vasculature, reducing innervation density in the heart. Tipping the balance towards chemo-repulsive Sema3a and inducing denervation increases the risk of arrhythmias in the aging heart, which is reversible with senolytic drugs (*Wagner et al., 2023*). This shows that even after birth, maintaining the balance between chemo-attraction and repulsion remains important throughout life and is targetable with therapeutics.

Only a relatively small number of attractive or repulsive axon guidance cues are known, and while we now know the involvement of key guidance factors in the heart, knowledge on one major player is still lacking. Here, we report a crucial novel role for the Slit-Robo signalling pathway in cardiac innervation guidance during development. The Roundabout (Robo) transmembrane receptors and their Slit ligands were first studied for their role in axonal guidance in the embryonic nervous system (*Blockus and Chédotal, 2016*). However, many new roles have been identified since, especially in cancer and embryogenesis (*Blockus and Chédotal, 2016*; *Jiang et al., 2019*). In the mammalian heart, the pathway is involved in cardiac chamber formation, cardiac neural crest migration and adhesion, the development of the pericardium and caval veins, membranous ventricular septum, and valve formation (*Zhao and Mommersteeg, 2018*; *Zhao et al., 2022*; *Mommersteeg et al., 2015*; *Mommersteeg et al., 2013*). Here, using constitutive and conditional knock-out approaches, axon guidance experiments and functional analysis, we show that interaction of Slit2 and Robo1 is important for correct innervation patterning during heart development. This knowledge will help us better understand the mechanisms underlying dysregulation of innervation during development, aging and disease, and how to promote appropriate nerve growth.

## Results

### Constitutive knock-out analysis indicates a role for Slit2 and Robo1 during cardiac innervation development

To test if Slit-Robo signalling is important for cardiac innervation guidance, we first analysed constitutive mouse knock-outs for ventricular innervation at embryonic day (E)14.5/E15.5 and E18.5 using Peripherin to label the total innervation (*Figure 1A–B*). *Slit1*[−/−] hearts did not show a phenotype at E14.5, but *Slit2*[−/−] hearts showed significantly reduced ventricular innervation. While *Slit3*[−/−] hearts showed increased levels of innervation compared to wild-type littermate controls at E14.5, this difference was not significant anymore at E18.5. Whereas the innervation was not affected in *Robo2*[−/−] mutants, the phenotype seen in *Slit2*[−/−] hearts was recapitulated in *Robo1*[−/−] hearts at E14.5 and still present at E18.5. Simultaneously knocking out *Robo1* and *Robo2* did not further increase the defects seen in the single *Robo1*[−/−] hearts at E15.5. We confirmed that these results were not influenced by differences in ventricular volume between the mutants and wild-type controls (*Figure 1C*). This data indicates a role for Slit2 and Robo1 during cardiac innervation development.

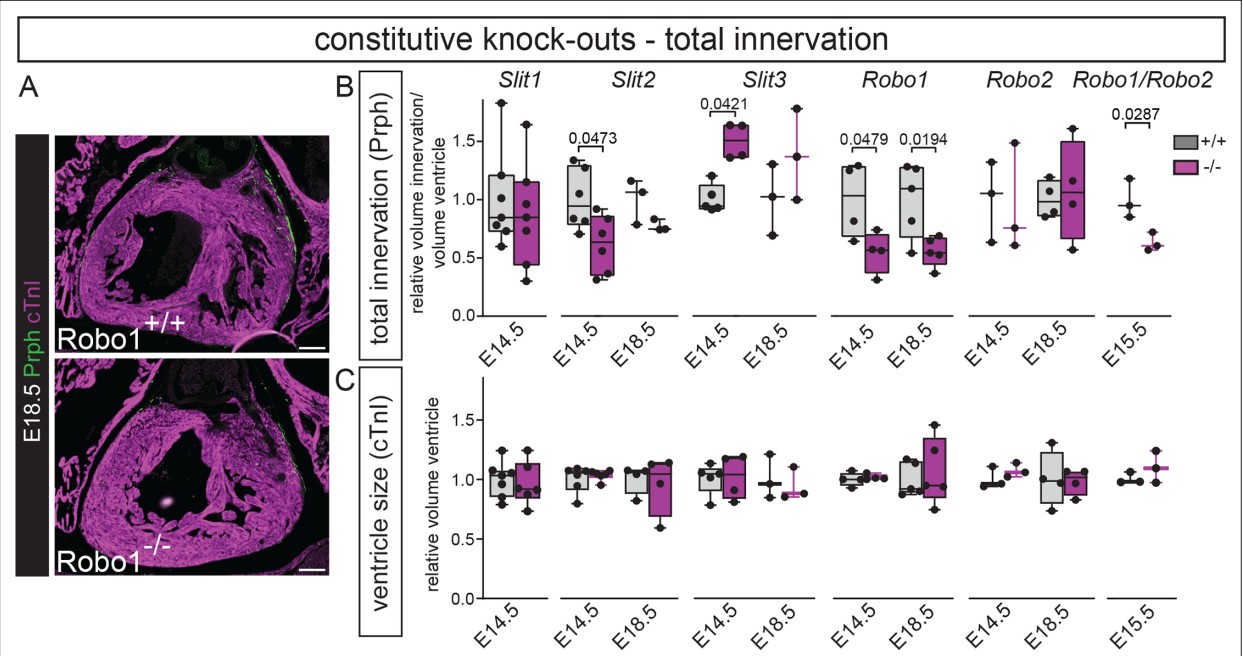

**Figure 1.** Constitutive knock-out analysis indicates a role for Slit2, Slit3, and Robo1 during cardiac innervation development. (**A**) Representative images of *Robo1⁺/⁺* and *Robo1⁻/⁻* sections stained by immunohistochemistry for the myocardium using cardiac Troponin I (cTnI) in purple and total innervation using Peripherin (Prph) in green. Arrowhead points to reduced total innervation staining in green in the *Robo1⁻/⁻*. (**B**) Quantification of total ventricular innervation corrected for ventricle volume (myocardial marker cTnI) in constitutive knock-outs and wild-type controls. T-test or one-way ANOVA with Tukey's test per mouse line. (**C**) Quantification of total ventricular volume based on myocardial marker cTnI in constitutive knock-outs and wild-type controls. T-test or one-way ANOVA with Tukey's test per mouse line. Scale bars, 100 µm.

The online version of this article includes the following source data for figure 1:

**Source data 1.** Source data *Figure 1B* - total innervation measurements.

**Source data 2.** Source data *Figure 1C* - ventricle size measurements.

## The coronary endothelium acts as a source of chemo-attractive Slit2 that guides the Robo1-positive nerves during heart development

Previously, we have shown expression of the different Slit ligands and Robo receptors in cell types, such as the myocardium, endocardium, and cushions/valves and their involvement in congenital heart disease (*Zhao et al., 2022*; *Mommersteeg et al., 2015*; *Mommersteeg et al., 2013*). As cardiac innervation follows the developing coronary vasculature (*Salamon et al., 2023*; *Poltavski et al., 2019*), here, we specifically focused on expression of the genes in the ventricular innervation and coronaries (*Figure 2A*). Immunohistochemistry at E18.5 found Slit2 ligand in the endothelium of the coronary veins and arteries, but not in the nerves extending into the heart. At E14.5, Slit3 was neither present in the vessels nor innervation (data not shown), but at E18.5, Slit3 was highly present in the coronary smooth muscle wall and in the larger epicardial nerves, and minimally in intra-myocardial nerves. Both the Robo1 and Robo2 receptors were present in the nerves and at low levels in the arteries. Together with the knock-out data, this suggests that the coronary endothelium could act as a source of chemo-attractive Slit2 that guides the Robo1-positive nerves during heart development.

Therefore, we further focused on the role of Slit2 and knocked out *Slit2* specifically in the endothelium, which recapitulated the defects seen in the constitutive *Slit2⁻/⁻* hearts (*Figure 2B*), which were still visible at E18.5. As Slit2 is also highly expressed in trabecular myocardium, which could act as long-range guidance cue, we additionally removed *Slit2* specifically from the myocardium, which did not show a phenotype compared to wild-type controls at E18.5. While the constitutive double *Robo1/2* knockout did not show an additional role for Robo2 over Robo1 (*Figure 1B*), this could be caused by Robo2 being expressed in both the endothelium and innervation resulting in opposing signals. We hence deleted *Robo2* in the innervation and endothelium of constitutive *Robo1⁻/⁻* mice. *Robo1⁻/⁻* hearts

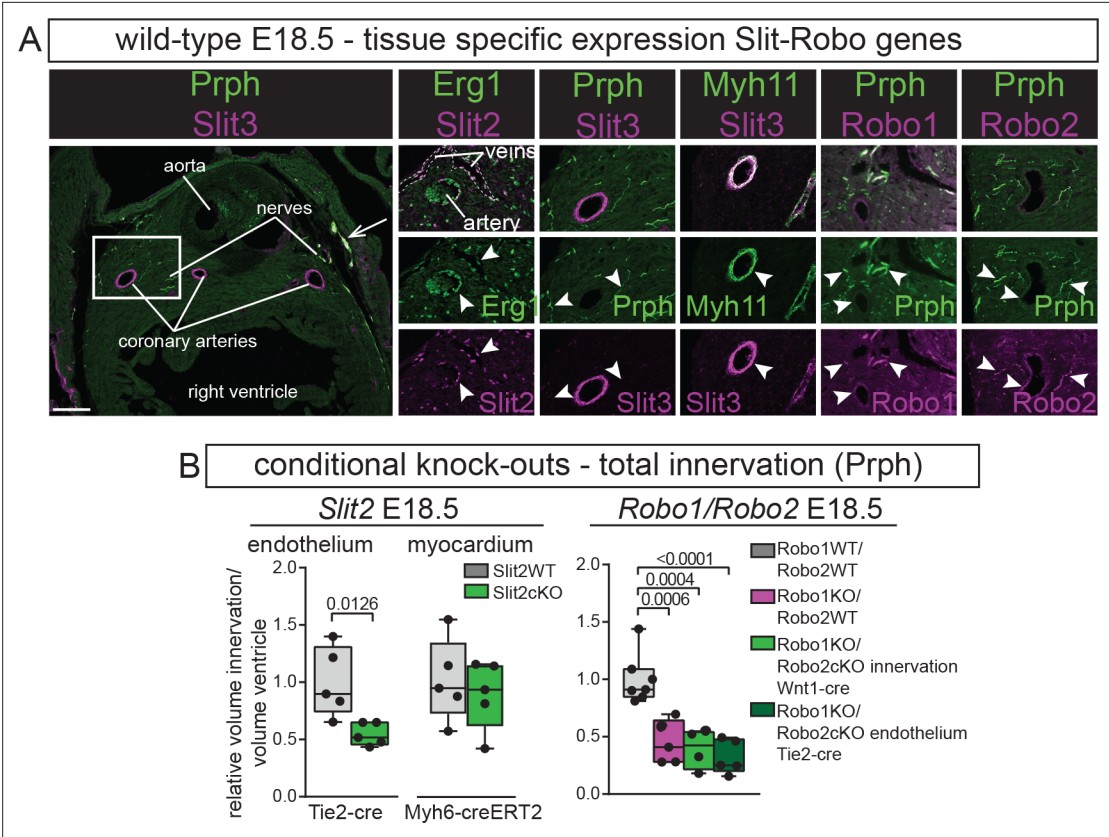

**Figure 2.** The coronary endothelium acts as a source of chemo-attractive Slit2 that guides the Robo1-positive nerves during heart development. (**A**) Immunohistochemistry for Slit2, Slit3, Robo1, Robo2, endothelial marker Erg1, innervation marker Peripherin (Prph), and smooth muscle marker Myh11 at E18.5. Box indicates region of magnification. Arrow points to overlap Slit3 and Prph in the large epicardial nerves. White arrowheads point to region of absence or presence of co-localisation of the indicated genes. (**B**) Measurements of total ventricular innervation corrected for ventricle volume in the indicated conditional knock-outs and wild-type controls. T-test or one-way ANOVA with Tukey's test. Scale bar, 100 μm.

The online version of this article includes the following source data for figure 2:

**Source data 1.** Source data *Figure 2B* - total innervation measurements.

showed the same reduction in innervation compared to wild-types as before, which was again not enhanced by specific removal of *Robo2* from the innervation or endothelium (*Figure 2B*).

## Reduced sympathetic innervation in the absence of endothelial Slit2 limits sympathetic stimulation of heart rate

To characterize the type of innervation affected in the endothelial Slit2 knock-out, we specifically quantified sympathetic innervation, which recapitulated the defects seen in the total innervation analysis (*Figure 3A–B*), showing that endothelial Slit2 guides the sympathetic innervation during heart development. To ensure that endothelial knock-out of Slit2 does not affect coronary vessel formation, we also quantified coronary vessel volumes, showing no difference to controls (*Figure 3C–D*). Additionally, we validated the specificity of *Slit2* knock-out. Endothelial mutants showed absence of Slit2 from the endothelial cells; however, endocardial expression surrounding the trabeculae was still present (*Figure 3E*). This is likely caused by the secretion of Slit2 from the strongly Slit2-positive trabecular myocardium (*Mommersteeg et al., 2015*; *Mommersteeg et al., 2013*). To further confirm the chemo-attractive role of endothelial Slit2, we cultured wild-type stellate ganglia on their own or in the presence of wild-type or endothelial *Slit2* knock-out ventricles. Attraction of axons extending from the ganglia was strongly reduced in the absence of endothelial Slit2 compared to wild-type ventricles (*Figure 3F–G*). Finally, to confirm functional relevance, we compared heart rate in adult endothelial Slit2 mutants and controls. While baseline heart rates were similar, Slit2 knock-out reduced heart rate increase after sympathetic stimulation with isoproterenol (*Figure 3H–I*). Combined, this data shows

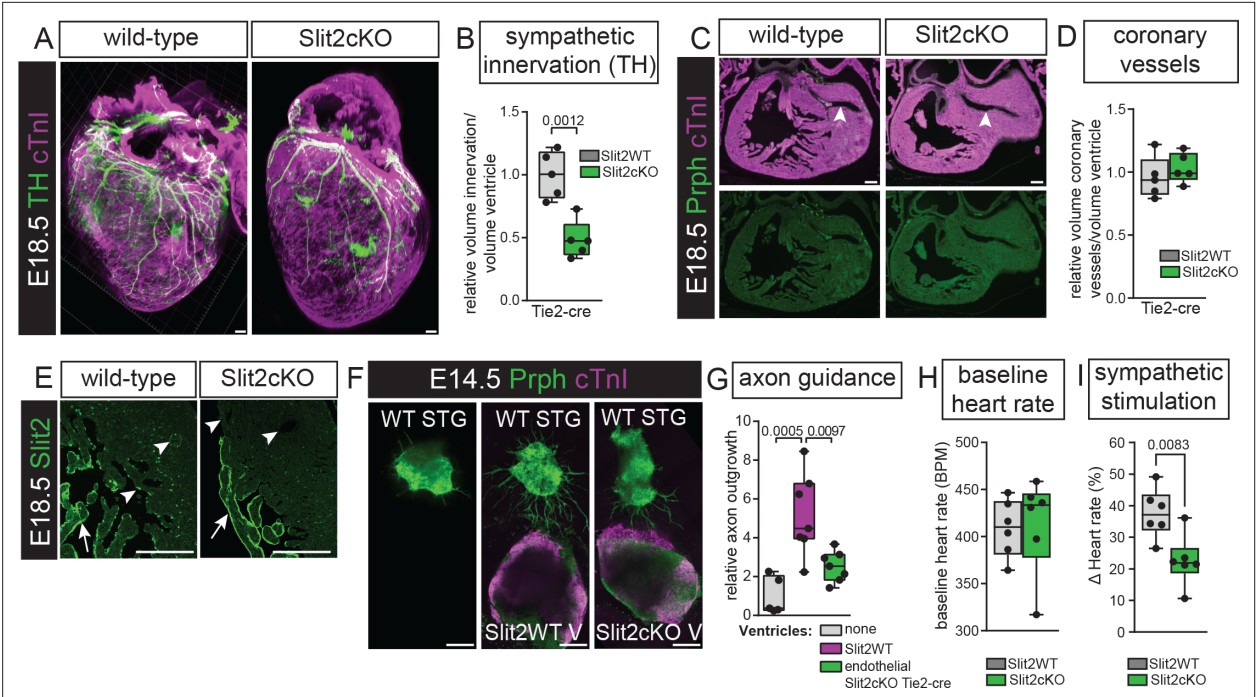

**Figure 3.** Reduced sympathetic innervation in the absence of endothelial Slit2 limits sympathetic stimulation of heart rate. (**A**) Immunohistochemistry for sympathetic innervation marker Tyrosine Hydroxylase (TH) and myocardial marker cardiac Troponin I (cTnI) on whole mount E18.5 hearts. (**B**) Quantification of sympathetic innervation corrected for ventricle volume in endothelial Slit2 knock-outs versus controls. T-test. (**C**) Immunohistochemistry for total innervation marker Peripherin (Prph) and myocardial marker cTnI on E18.5 sections. Arrowheads point to the normal development of the coronary vessels. (**D**) Quantification of coronary vessel volume corrected for ventricle volume in endothelial Slit2 knock-outs versus controls. T-test. (**E**) Immunohistochemistry for Slit2 on endothelial Slit2 knock-outs versus controls, showing the lack of Slit2 in the endothelium. (**F**) E14.5 ventricles and stellate ganglia (STG) cultured in three-dimensional collagen matrix (Chemicon) for 3 days. Immunohistochemistry for Prph and cTnI. (**G**) Quantification of axon outgrowth. One-way ANOVA with Tukey's Test. (**H**) Quantification of baseline heart rate using an electrocardiogram. (**G**) Quantification of heart rate after stimulations with isoproterenol. Scale bar, 100 μm.

The online version of this article includes the following source data for figure 3:

**Source data 1.** Source data *Figure 3B* - sympathetic innervation measurements.

**Source data 2.** Source data *Figure 3D* - coronary vessel measurements.

**Source data 3.** Source data *Figure 3G* - axon guidance measurements.

**Source data 4.** Source data *Figure 3H* - baseline heart rate measurements.

**Source data 5.** Source data *Figure 3I* - sympathetic stimulation measurements.

that endothelial Slit2 guides sympathetic innervation during heart development and absence results in functional defects.

## Discussion

We have identified an important new chemoattractive pathway mediated by Slit2 which is required for correct cardiac innervation development. This is somewhat unexpected as Slit2 is mainly known as a chemo-repulsive ligand, repelling extending axons (*Plump et al., 2002*). However, many chemo-active ligands are bifunctional and can promote as well as inhibit axon growth, depending on the circumstances (*Wong et al., 1999*; *de Castro et al., 1999*; *Colamarino and Tessier-Lavigne, 1995*). In the case of Slit2, it has been shown that concentration levels are important for its bifunctionality as well as proteolytic cleavage, with possible specific biological functions of its N- and C-terminal fragments which are distinct from the full-length protein (*Pilling et al., 2019*; *Bhosle et al., 2023*; *Nguyen-Ba-Charvet et al., 2001*). Additionally, the specific chemo-active function of Slit2 can further be modulated by the type of extracellular matrix present (*Nguyen-Ba-Charvet et al., 2001*). How the attractive function of Slit2 in axon guidance during heart development is regulated will need to be determined.

As the endothelial-specific deletion of Slit2 affects endothelial cells throughout the body, the axons extending into the heart could also be influenced before reaching the heart.

The increase in innervation in constitutive *Slit3*$^{-/-}$ hearts at E14.5 could suggest that Slit3 has a chemo-repulsive role during early innervation development. However, this effect was not visible anymore around birth and could be caused by the nerves following the abnormal caval vein connections to the heart that are present in this mutant (*Mommersteeg et al., 2013*). An indirect effect on innervation is supported by the limited expression of Slit3 in relevant cell types at E14.5. After E14.5, multiple cell types start to express Slit3 and the gene also becomes expressed in the large innervation, indicating expression is only turned on as the axons mature. The Slit-Robo pathway is best known for its role during nervous system development, where commissural neurons only become sensitive to repulsive Slit after midline crossing to prevent them from recrossing (*Bisiak and McCarthy, 2020*). A similar temporal activation of the different genes of the pathway in the different cell types would allow for complex control over where and when axons develop, which will need further investigation. While our data indicates that the innervation responds to endothelial Slit2 through the Robo1 receptor, we also find Robo1 expressed in the endothelium itself. This needs further examination using a conditional Robo1 line, which we failed to generate successfully (*Zhao et al., 2022*). These complicated spatio-temporal expression patterns of the genes of the Slit-Robo pathway could be key for determining the sympathetic versus parasympathetic balance or the patterning along veins and arteries (*Poltavski et al., 2019*; *Schneider et al., 2012*). Thus, our discovery of an important role for the Slit-Robo signalling pathway during cardiac innervation development raises new exciting questions that will need further investigation.

## Materials and methods
### Mouse lines
All experiments were performed in accordance with the UK Animals (Scientific Procedures) Act 1986 and institutional guidelines. *Robo1*$^{tm1Wia}$ (*Andrews et al., 2008*) and *Robo2*$^{tm1Rilm}$ (*Lu et al., 2007*) were obtained from William Andrews (UCL, London, UK). *Robo1*$^{tm1Matl}$; *Robo2*$^{tm1Rilm}$ (*Rama et al., 2015*), Slit1$^{tm1Matl}$ and Slit2$^{tm1Matl}$ (*Plump et al., 2002*), *Slit2*$^{tm1.1lcs}$ (*Gibson et al., 2014*) from Alain Chedotal (Institut de la Vision, Paris, France). Slit3$^{tm1.1Dor}$ (*Yuan et al., 2003*) from David Ornitz (*Jiang et al., 2019*). *Tg(Wnt1-cre)*$^{11Rth}$ (*Danielian et al., 1998*), *Tg(Tek-cre)*$^{12Flv}$ (*Tie2-cre*) (*Koni et al., 2001*) and *Tg(Myh6-cre/Esr1\*)*$^{1Jmk}$ (*Sohal et al., 2001*) were obtained from the Jackson lab. All lines were maintained on a C57/BL6J background (C57BL/6JOlaHsd, model code 057, Envigo). The day the vaginal plug was found was considered as embryonic day (E) 0.5. Cre allele *Tg(Myh6-cre/Esr1\*)*$^{1Jmk}$ was induced by oral gavage with five consecutive doses of 20 mg/kg 4-hydroxytamoxifen (Sigma, T5648) at E12.5 to E16.5.

### Tissue processing and immunohistochemistry-paraffin section staining
Fluorescent immunohistochemistry was performed as previously described (*Mommersteeg et al., 2010*). Embryos were dissected and fixed overnight in 4% paraformaldehyde (ChemCruz, SC-281692). Following a brief wash in PBS (Gibco, 18912–014), the samples were dehydrated using an ethanol gradient (2 hr incubations in 70%, 80%, 90%, and 96% and two 1 hr incubations in 100% ethanol, Sigma-Aldrich, 32221–2.5 L-M) before being incubated in 1-butanol (MP Biomedicals, 194001) at room temperature overnight. Before mounting the embryos in sectioning moulds, three 2 hr incubations were performed in paraffin (Paraplast plus, Sigma-Aldrich, P3683) at 65 °C. Using a Leica microtome (Microm HM325), 10 µm sections were collected, mounted on Superfrost Plus glass slides (VWR, 631–0108) ensuring even coverage of the heart (one in every three sections per staining combination) and were allowed to dry overnight at 33 °C.

The slides were de-waxed in Xylene (Fisher Scientific, X/0200/17) twice for 5 min and rehydrated in an ethanol gradient (100%, 100%, 96%, 90%, 80%, 70%) for 1 min each before being incubated for one more minute in PBS. The samples were then boiled for 4 min in a pressure cooker in antigen-unmasking solution (H-3300, Vector Laboratories) and were allowed to cool down in PBS-T (Tween-20 0.1%, Chemcruz, sc-29113) before a ring was drawn around them using an ImmEdge pen (Vector Laboratories, H4000). The sections were blocked for 30 min at room temperature in a humidified chamber using blocking reagent [0.5% blocking reagent powder (Akoya Bioscience,

FP1012), 0.15 M NaCl, 0.1 M Tris-HCl (pH 7.5)] and were then incubated with primary antibody diluted in blocking solution overnight at room temperature. Following three 5 min washes in PBS-T, the secondary antibody was applied (1:200 in blocking solution, Invitrogen, Alexa range) for 2 hr at room temperature.

For the Slit and Robo antibodies, an additional amplification step was added to enhance the signal using the TSA kit (NEL756001KT, Perkin and Elmer). Instead of an Alexa secondary antibody, a biotinylated secondary antibody was used at a 1:200 dilution in blocking solution for 45 min at room temperature, followed by three 5 min washes in PBS-T prior to a 30 min incubation with conjugated Streptavidin-Horse Radish Peroxidase (Vector Laboratories, SA-5004) and then three 5 min washes in PBS-T. Either fluorescein or tetramethylrhodamine (in DMSO, NEL756001KT) diluted at 1:100 in amplification buffer was then added to the sections for 3 min, followed by three 5 min washes with PBS and staining with DAPI (2.5 µg/ml, Sigma, MBD0015). Mowiol 4–88 (Applichem, A9011,1000) was used to mount the slides, which were cured at 37 °C in the dark. Primary antibodies against the following proteins were used: goat polyclonal anti-cardiac Troponin I (cTnI, 1:200; Hytest Ltd, 4T21/2), mouse monoclonal anti-Myosin heavy chain (MF20, 1:50, HSHB, AB-2147781), rabbit polyclonal anti-Peripherin (Prph, 1:200; Millipore, AB1530), rabbit polyclonal anti-Tyrosine Hydroxylase (TH, 1:200; Millipore, AB152), goat polyclonal anti-Robo1 (1:200; R&D Systems, AF1749), goat polyclonal anti-Robo2 (1:200; R&D Systems, AF3147), sheep polyclonal anti-Slit2 (1:200; R&D Systems, AF5444), rat monoclonal anti-Slit3 (1:200; R&D Systems, MAB3629), rabbit polyclonal anti-Erg1 (1:200, Abcam, ab110639), and rabbit polyclonal anti-Smooth muscle myosin heavy chain 11 (Myh11, 1:200, Abcam, ab125884). Imaging was performed on a Nikon microscope (Nikon, Eclipse Ci-L). Images were processed using ImageJ to generate magenta and green colour combinations (*Schneider et al., 2012*).

## Volume measurements

To quantitate the nerve density in the ventricle, every third transverse 10 µm section capturing the entire diameter of the heart was stained as detailed above for cTnI and Peripherin or TH and imaged using a Nikon microscope (Nikon, Eclipse Ci-L). The images were loaded into 3D reconstruction software (Amira 6.3.0, Thermo Fisher Scientific). Images were resampled, aligned, and different labels were created for the myocardium of the ventricle based on cTnI thresholding and for the innervation surrounding the ventricle, based on Peripherin thresholding for total innervation or based on TH thresholding for sympathetic innervation. All ventricular innervation was included, epicardial and intramyocardial. The label data was used to generate volume information for the myocardium of the ventricle and innervation. To measure the coronary vessels, similar volume measurements were performed for the lumen of the coronary vasculature. Sample sizes were determined based on previous experiences with these measurements. Measurements were performed blind and no animals were excluded. For controls, we used Cre-negative hetero- and homozygous floxed littermates as well as Cre-positive mice that had not inherited floxed alleles.

## Immunohistochemistry-whole mount staining

Mouse hearts were prepared and fixed as described above and subsequently dehydrated by methanol series (50–100%) and bleached overnight in 6% $H_2O_2$ in methanol. Hearts were then rehydrated, blocked in PBSGT blocking buffer (0.2% gelatin (Sigma, G7041), 0.5% Triton X-100 (Merck, T8787), and 0.01% thimerosal in PBS) for 2–4 days. Primary antibodies were diluted in PBSGT as follows: goat polyclonal anti-cardiac Troponin I (cTnI, 1:200; Hytest Ltd, 4T21/2), rabbit polyclonal anti-Tyrosine Hydroxylase (TH, 1:200; Millipore, AB152). Hearts were incubated in primary antibodies at 37 °C for 1 week. Afterwards, the hearts were washed six times with PBSGT at room temperature before secondary antibody incubation in PBSGT for 2 days. Subsequently, the hearts were washed six times for 1 hr in PBSGT. The hearts were then dehydrated in a graded tetrahydroflurane (THF, Sigma-Aldrich, 186562) series (50%, 80%, and 100%) and delipidated in dichloromethane (DCM, Sigma-Aldrich, D65100). Finally, samples were incubated in dibenzyl ether (DBE, Sigma-Aldrich, 108014) until clear and stored in DBE at room temperature. Samples were imaged on a light-sheet microscope (Zeiss, lightsheet Z.1). 3D-rendered images were visualized and captured with Imaris software (Bitplane, Version 7.6.4).

## Explant co-cultures

Ventricles and stellate ganglia (STG) were dissected out from E14.5 or E15.5 mouse embryos with the indicated genotype, placed in PBS, and cut into pieces and plated into a three-dimensional collagen matrix (3D collagen cell culture system, Chemicon) concurrently in DMEM media containing 15% fetal bovine serum (FBS, Sigma, F7524) for three days. The ventricle was placed 300–500 µm away from the STG. Myocardium and neurite outgrowth were visualised by immunostaining whole-mount explants with rabbit polyclonal anti-Peripherin (1:200; Millipore, AB1530) and goat polyclonal anti-cardiac Troponin I (cTnI, 1:200; Hytest Ltd, 4T21/2), followed by immunofluorescence detection using Alexa Fluor secondary antibodies. For quantitative analysis of the extent of axon growth from the STG into the heart, fluorescence images were captured at 4 µm steps by using a spinning disk confocal microscope (PE Ultraview spinning disk). These pictures were projected over the Z axis and then merged manually using Volocity software (Quorum Technologies Inc). STG explants were divided into forward and backward portions from the centre. The axons growing toward the ventricles were measured using 3D reconstruction software (Amira 6.3.0, Thermo Fisher Scientific).

## Electrocardiography (ECG) studies

ECG recording of heart rate (HR) was acquired with MouseMonitor software (INDUS instruments). 2–3 months old male mice were anaesthetized with 1.5% isoflurane (IsoFlo Zoetis) in oxygen and kept at a constant 37°C temperature using a heating pad for the duration of analysis. Following baseline reading, mice were challenged by isoproterenol (Acros, 14190250, 0.5 mg/kg by intraperitoneal injection).

## Statistics

Detailed statistics and p-values are indicated in the figure legends. T-test or one-way ANOVA with Tukey's Test have been applied as indicated. Minimum to maximum box-violin plots are shown with all data points.

## Acknowledgements

We would like to thank the staff at our animal facility at the University of Oxford.

This work was supported by the British Heart Foundation (PG/15/50/31594), EMBO (ALTF 441–2010), Netherlands Organisation for Scientific Research (825.10.025), and European Research Council (ERC) (715895, CAVEHEART) to MTMM, Oxford BHF Centre of Research Excellence by grant code (RE/18/3/34214) to SB, UKRI-MRC (MR/W006731/1) to KL, BHF FS/19/31/34158 to JV, and BBSRC BB/W015277/1 to JV and MTMM.

## Additional information

### Funding

| Funder | Grant reference number | Author |
| --- | --- | --- |
| British Heart Foundation | PG/15/50/31594 | Mathilda Mommersteeg |
| European Molecular Biology Organization | ALTF 441-2010 | Mathilda Mommersteeg |
| ZonMw | 825.10.025 | Mathilda Mommersteeg |
| European Research Council | 715895 | Mathilda Mommersteeg |
| European Research Council | CAVEHEART | Mathilda Mommersteeg |
| British Heart Foundation | RE/18/3/34214 | Susann Bruche Mathilda Mommersteeg |
| Medical Research Council | MR/W006731/1 | Konstantinos Lekkos |

| Funder | Grant reference number | Author |
|---|---|---|
| British Heart Foundation | FS/19/31/34158 | Joaquim Miguel Vieira |
| Biotechnology and Biological Sciences Research Council | BB/W015277/1 | Joaquim Miguel Vieira Mathilda Mommersteeg |

The funders had no role in study design, data collection and interpretation, or the decision to submit the work for publication.

## Author contributions

Juanjuan Zhao, Formal analysis, Investigation, Methodology, Writing – review and editing; Susann Bruche, Formal analysis, Investigation, Methodology, Writing – original draft, Writing – review and editing; Konstantinos Lekkos, William D Andrews, Investigation, Writing – review and editing; Carolyn Carr, Methodology, Writing – review and editing; Joaquim Miguel Vieira, Writing – review and editing; John Parnavelas, Supervision, Writing – review and editing; Mathilda Mommersteeg, Conceptualization, Formal analysis, Supervision, Funding acquisition, Investigation, Methodology, Writing – original draft, Project administration, Writing – review and editing

## Author ORCIDs

Susann Bruche ⓘ https://orcid.org/0000-0002-5814-7166
Konstantinos Lekkos ⓘ https://orcid.org/0000-0001-5044-2876
Joaquim Miguel Vieira ⓘ https://orcid.org/0000-0003-2023-6304
Mathilda Mommersteeg ⓘ https://orcid.org/0000-0001-6624-3038

## Ethics

All experiments were carried out under appropriate Home Office licenses and in compliance with the revised Animals (Scientific Procedures) Act 1986 in the UK, and all have been approved by Oxford's central Committee on Animal Care and Ethical Review (ACER).

Reviewer #1 (Public review): https://doi.org/10.7554/eLife.105932.3.sa1
Reviewer #2 (Public review): https://doi.org/10.7554/eLife.105932.3.sa2
Author response https://doi.org/10.7554/eLife.105932.3.sa3

# Additional files

## Supplementary files

MDAR checklist

## Data availability

The source data file contains the numerical data used to generate the figures.

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
