## [Editor Report · eLife Assessment]

This study presents **important** findings on the role of Slit-Robo signaling in cardiac innervation. The evidence supporting the main claims of the authors is **convincing**. The use of several mouse models including constitutive and cell type specific knockout models make the findings more robust. The scope of the presented studies is fitting, as they primarily focus on evaluating the phenotypic changes in cardiac innervation following the loss of various Slit or Robo genes

---

## [Referee Report · Reviewer #1 (Public review)]

The study aims to determine the role of Slit-Robo signaling in the development and patterning of cardiac innervation, a key process in heart development. Despite the well-studied roles of Slit axon guidance molecules in the development of the central nervous system, their roles in the peripheral nervous system are less clear. Thus, the present study addresses an important question. The study uses genetic knockout models to investigate how Slit2, Slit3, Robo1, and Robo2 contribute to cardiac innervation

Using constitutive and cell type-specific knockout mouse models, they show that the loss of endothelial-derived Slit2 reduces cardiac innervation. Additionally, Robo1 knockout, but not Robo2 knockout, recapitulated the Slit2 knockout effect on cardiac innervation, leading to the conclusion that Slit2-Robo1 signaling drives sympathetic innervation in the heart. Finally, the authors also show a reduction in isoproterenol-stimulated heart rate but not basal heart rate in the absence of endothelial Slit2.

The conclusions of this paper are mostly well supported by the data, but there are several limitations:

(1) It is well established that Slit ligands undergo proteolytic cleavage, generating N- and C-terminal fragments with distinct biological functions. Full-length Slit proteins and their fragments differ in cell association, with the N-terminal fragment typically remaining membrane-bound, while the C-terminal fragment is more diffusible. This distinction is crucial when evaluating the role of Slit proteins secreted by different cell types in the heart. However, this study does not examine or discuss the specific contributions of different Slit2 fragments, limiting its mechanistic insight into how Slit2 regulates cardiac innervation. While these points are mentioned in the discussion, they are not incorporated into the interpretation of the data or the presented model.

(2) The endothelial-specific deletion of Slit2 leads to its loss in endothelial cells across various organs and tissues in the developing embryo. Therefore, the phenotypes observed in the heart may be influenced by defects in other parts of the embryo, such as the CNS or sympathetic ganglia, and this possibility cannot be ruled out. The data presented in the manuscript does not dissect the relative contributions of endothelial Slit2 loss in the heart versus secondary effects arising from other organ systems. Without tissue-specific rescue or complementary conditional models, it remains unclear whether the observed cardiac phenotypes are a direct consequence of local endothelial Slit2 deficiency or an indirect outcome of broader developmental perturbations.

---

## [Referee Report · Reviewer #2 (Public review)]

The aims of investigating Slit-Robo signaling in cardiac innervation were achieved by the experiments designed. The authors demonstrate that endothelial Slit2 signaling through Robo1 drives sympathetic innervation. While questions remain regarding signal regulation and interplay between established axon guidance signals and the further role of other Slit ligands and Robo expression in endothelium, the results strongly support the conclusions drawn.

Writing and presentation are easy to follow and well structured. Appropriate controls are used, statistical analysis applied appropriately, and experiments directly test aims following a logical story.

The authors demonstrate a novel mechanism for Slit-Robo signaling in cardiac sympathetic innervation. The data establishes a framework for future studies.

The authors have updated their discussion to highlight the need for investigation of the role of proteolytic cleavage of Slit2 as well as the potential for defects in other tissues due to endothelial knockout of Slit2 influencing cardiac innervation.

---

## [Author Response]

The following is the authors’ response to the original reviews

**Public Reviews:**

**Reviewer #1 (Public review):**
The study aims to determine the role of Slit-Robo signaling in the development and patterning of cardiac innervation, a key process in heart development. Despite the well-studied roles of Slit axon guidance molecules in the development of the central nervous system, their roles in the peripheral nervous system are less clear. Thus, the present study addresses an important question. The study uses genetic knockout models to investigate how Slit2, Slit3, Robo1, and Robo2 contribute to cardiac innervation.Using constitutive and cell type-specific knockout mouse models, they show that the loss of endothelial-derived Slit2 reduces cardiac innervation. Additionally, Robo1 knockout, but not Robo2 knockout, recapitulated the Slit2 knockout effect on cardiac innervation, leading to the conclusion that Slit2-Robo1 signaling drives sympathetic innervation in the heart. Finally, the authors also show a reduction in isoproterenol-stimulated heart rate but not basal heart rate in the absence of endothelial Slit2.The conclusions of this paper are mostly well supported by the data, but some should be modified to account for the study's limitations and discussed in the context of previous literature.

We would like to thank the reviewer for their positive evaluation of our manuscript and in response to the reviewer’s comments we have extended the discussion as indicated below.

(1) It is well established that Slit ligands undergo proteolytic cleavage, generating N- and C-terminal fragments with distinct biological functions. Full-length Slit proteins and their fragments differ in cell association, with the N-terminal fragment typically remaining membrane-bound, while the C-terminal fragment is more diffusible. This distinction is crucial when evaluating the role of Slit proteins secreted by different cell types in the heart. However, this study does not examine or discuss the specific contributions of different Slit2 fragments, limiting its mechanistic insight into how Slit2 regulates cardiac innervation.

This is a valid point and it will be of interest for future studies to investigate the specific effects of the full length versus N- and C-terminal fragments in the context of cardiac innervation development. We have updated our discussion with a clearer reference to the proteolytic cleavage of Slit2.

(2) The endothelial-specific deletion of Slit2 leads to its loss in endothelial cells across various organs and tissues in the developing embryo. Therefore, the phenotypes observed in the heart may be influenced by defects in other parts of the embryo, such as the CNS or sympathetic ganglia, and this possibility cannot be ruled out.

We agree and we have now added this possibility to the discussion.

**Reviewer #2 (Public review):**
The aims of investigating Slit-Robo signaling in cardiac innervation were achieved by the experiments designed. While questions remain regarding signal regulation and interplay between established axon guidance signals and further role of other Slit ligands and Robo expression in endothelium, the results strongly support the conclusions drawn.Writing and presentation are easy to follow and well structured, Appropriate controls are used, statistical analysis applied appropriately, and experiments directly test aims following a logical story.The authors demonstrate a novel mechanism for Slit-Robo signaling in cardiac sympathetic innervation. The data establishes a framework for future studies.

We would like to thank the reviewer for these positive comments.

Recommendations:Further assessment of interplay between Slit ligands as well as other signaling pathways (Semaphorin, NGF, etc) could be investigated. Is it possible to rescue the phenotype by modulation of other signaling pathways? Can combined Slit2/Slit3 KO rescue? Additionally, as the authors state, conditional Robo1 knockouts will be important to validate the findings of constitutive knockout.

Our study has provided the first data on the role of Slit-Robo signalling during cardiac innervation development and a base for exploring the interesting further questions the reviewer raises.

**Recommendations for the authors:**

**Reviewer #1 (Recommendations for the authors):**
There is a typo on line 83 (disease).

This has been corrected.